

# Efficient algorithm for directed text detection based on rotation decoupled bounding box

Songma Wei[1], Minrui Lu[2], Bingsan Chen[1], Tengjian Zhang[2], Fujiang Zhang[1] and Xiaodong Peng[1]

[1] Fujian Key Laboratory of Intelligent Machining Technology and Equipment, Fujian University of Technology, Fuzhou, China
[2] Fujian Wuyi Leaf Tobacco Co., Ltd., Shaowu, China

## ABSTRACT

A more effective directed text detection algorithm is proposed for the problem of low accuracy in detecting text with multiple sources, dense distribution, large aspect ratio and arbitrary alignment direction in the industrial intelligence process. The algorithm is based on the YOLOv5 model architecture, inspired by the idea of DenseNet dense connection, a parallel cross-scale feature fusion method is proposed to overcome the problem of blurring the underlying feature semantic information and deep location information caused by the sequential stacking approach and to improve the multiscale feature information extraction capability. Furthermore, a rotational decoupling border detection module, which decouples the rotational bounding box into horizontal bounding box during positive sample matching, is provided, overcoming the angular instability in the process of matching the rotational bounding box with the horizontal anchor to obtain higher-quality regression samples and improve the precision of directed text detection. The MSRA-TD500 and ICDAR2015 datasets are used to evaluate the method, and results show that the algorithm measured precision and $F_1$-score of 89.2% and 88.1% on the MSRA-TD500 dataset, respectively, and accuracy and $F_1$-score of 90.6% and 89.3% on the ICDAR2015 dataset, respectively. The proposed algorithm has better competitive ability than the SOTA text detection algorithm.

## INTRODUCTION

The goal of text detection is to locate text areas in a given image, which is a prerequisite for many application media, including multimedia retrieval (*Ye et al., 2005*), industrial automation (*Liang, Doermann & Li, 2005*) and optical character recognition (OCR) (*Vinciarelli, 2002*) system applications. With the development of convolutional neural networks (CNNs) (*Wang et al., 2012*), the existing text detection models have achieved better results on ICDAR 2013 and COCO-Text real datasets, where the models usually employ horizontal bounding boxes (HBBs) (*Xia et al., 2018*) for target localisation. However, in most existing detection tasks, the target texts in a given image own the characteristics of dense distribution, a large aspect ratio, and arbitrary alignment directions.

Corresponding authors
Minrui Lu, 1352248635@qq.com
Bingsan Chen, bschen126@fjut.edu.cn

When the HBB model is used to detect the densely arranged directed text, the non-maximum suppression (NMS) (*Neubeck & Van Gool, 2006*) may remove some of the prediction bounding boxes due to the large intersection over union (IOU) (*Jiang et al., 2018*) between adjacent prediction boxes, eventually decreasing in the detection accuracy (*Liao et al., 2017*; *Bazazian et al., 2017*). Subsequently, some scholars proposed a rotating bounding box (OBB) (*Ma et al., 2018*; *Yang et al., 2021*) for target localisation, which can effectively separate densely arranged directed targets and reduce redundancy whilst decreasing the false kill of NMS on detected targets by adding angles, thus improving the accuracy of target localisation.

With the introduction of the OBB bounding box, the loss of the model at the boundary suddenly increases due to the existence of periodicity in the bounding box angle (*Yang et al., 2021*). As a result, the model cannot obtain the prediction results in the simplest and most direct approach. Scholars then proposed the APE (*Zhu, Du & Wu, 2020*) to represent the angle as a continuously varying periodic vector; Theyused a free detector with boundary discontinuity (CSL) (*Yang & Yan, 2020*) to transform the angle regression problem, and RSDet++ (*Qian et al., 2022*) to mitigate the angle loss discontinuity by modulating the bounding box rotation loss. Although the use of these methods can mitigate or even eliminate the effects caused by the angular periodicity of the bounding box, these approaches undoubtedly increase the computation of the model. In this regard, there is still a huge room for improvement in the field of target detection, oriented toward a series of optimization schemes for target detection.

This study proposes a text detection algorithm based on YOLOv5 model architecture to address the shortcomings of the preceding algorithms in dense distribution, large aspect ratio, and other directed text detection. A parallel cross-scale connection method is proposed in the feature fusion process to overcome the lack of underlying feature information and deep location information caused by the original sequential stacking method, thereby improving the multiscale feature fusion capability. An improved rotation decoupling bounding box detection module is also proposed and implemented. The OBB bounding box is decoupled into the HBB + angle combination method in the process of model training positive sample matching strategy to solve the rotation bounding box angle periodicity problem. As a result, the model detection precision is effectively improved, and the model parameter calculation is reduced.

The remainder of the article is organised as follows: first, the proposed method for directed text detection is introduced in the 'Proposed Methodology'. Then, the 'Analysis of Experimental Results' conducts experiments on the proposed method, and its effectiveness is analysed. Finally, the 'Conclusion and Future Work' summarises the proposed approach and provides an outlook for future work.

## PROPOSED METHOD

In this section, we initially illustrate the network architecture of the rotational decoupled oriented text detection algorithm. Then, we describe the proposed D-PANet feature fusion network module and the design of the rotational decoupling bounding box detection module in detail.

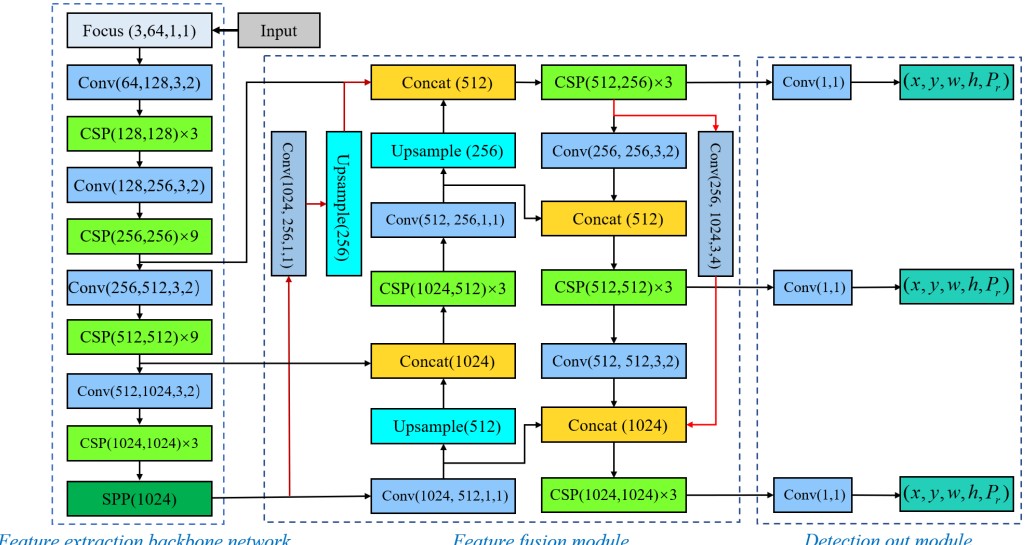

**Figure 1** Rotationally decoupled target text detection model.

## Network architecture

The proposed rotational decoupled oriented text detection algorithm is designed based on the YOLOv5 (*Zhu et al., 2021*) architecture, as shown in Fig. 1, the proposed architecture consists of four parts: input layer, feature extraction backbone network, feature fusion module and detection output module. Amongst them, the input layer and feature extraction backbone network remain unchanged from YOLOv5, whereas the feature fusion module and detection output module have been improved.

## D-PANet feature fusion network

Target detection in the feature fusion process heavily relies on the location information in low-level features and the semantic information in high-level features. Low-level features have fewer convolution times, resulting in relatively less semantic information, but the localisation information is accurate. By contrast, high-level features are rich in semantic information due to repeated convolutions, but the target localisation is more ambiguous. The D-PANet feature fusion network is used to effectively enhance the low-level feature semantic information and the high-level feature location informationin this article.

The proposed D-PANet model network is based on the idea of DenseNet (*Huang et al., 2017*) dense connectivity, which adds a parallel up-sampling and down-sampling channel to the multiscale features by using cross-scale connectivity, converting the feature fusion from the original sequential stacking method to a parallel two-branch structure. The structure can effectively enable each scale to acquire more features, enhance image feature fusion, and realise the improvement of semantic information and location information for each scale feature. The specific network of D-PANet is shown in Fig. 2.

Taking the acquisition of $F_5$ as an example, we retain the 1 * 1 convolutional layer that reduces the number of $C_5$ feature channels in the original YOLOv5 feature fusion

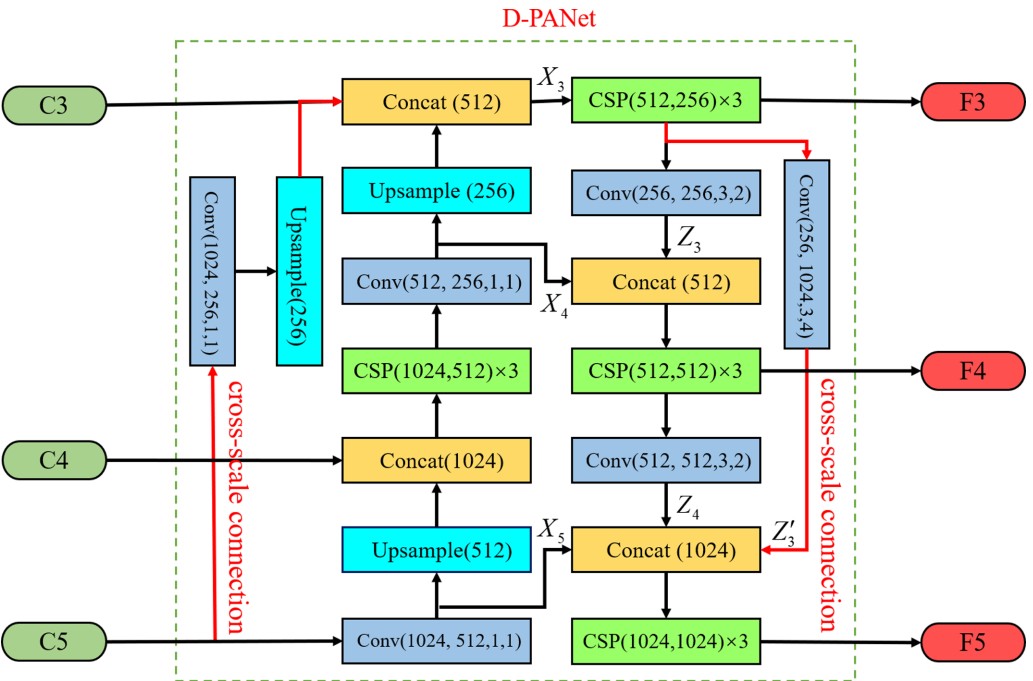

**Figure 2   D-PANet feature fusion network.**

module whilst retaining the down-sampling operation with a stride size of 2 for feature $F_4$. Then, the second branch is used to down-sample operations with a stride size of 4 for feature $F_3$. Finally, BottleneckCSP is used to eliminate the overlapping effect caused by the down-sampling process. The acquisition process is shown in Fig. 3, where each scale output $F_3$, $F_4$ and $F_5$ can be expressed as follows:

$$\begin{cases} F_3 = X_3 \\ F_4 = f(X_4, Z_3) \\ F_5 = f(X_5, Z_4, Z_3') \end{cases} \tag{1}$$

where $(X_3, X_4, X_5, Z_3, Z_3', Z_4)$ denotes the output features of each scale feature layer and $f$ denotes the *concat + conv* operation.

## Design and implementation of the rotational decoupling bounding box detection module

Figure 4 shows the output diagram of the rotational decoupled bounding box detection module. The image is passed through the feature extraction backbone network and D-PANet feature fusion module to obtain the final detection feature layer. The feature channel output dimension is 3(5+1), where 3 is the different preset aspect ratio anchor box, 5 indicates the detection output bounding box information, and 1 indicates the text confidence degree. In the training process of this detection module model, firstly, the rotationally decoupled bounding box representation is defined, and then the positive sample matching strategy between the rotationally decoupled bounding box and the

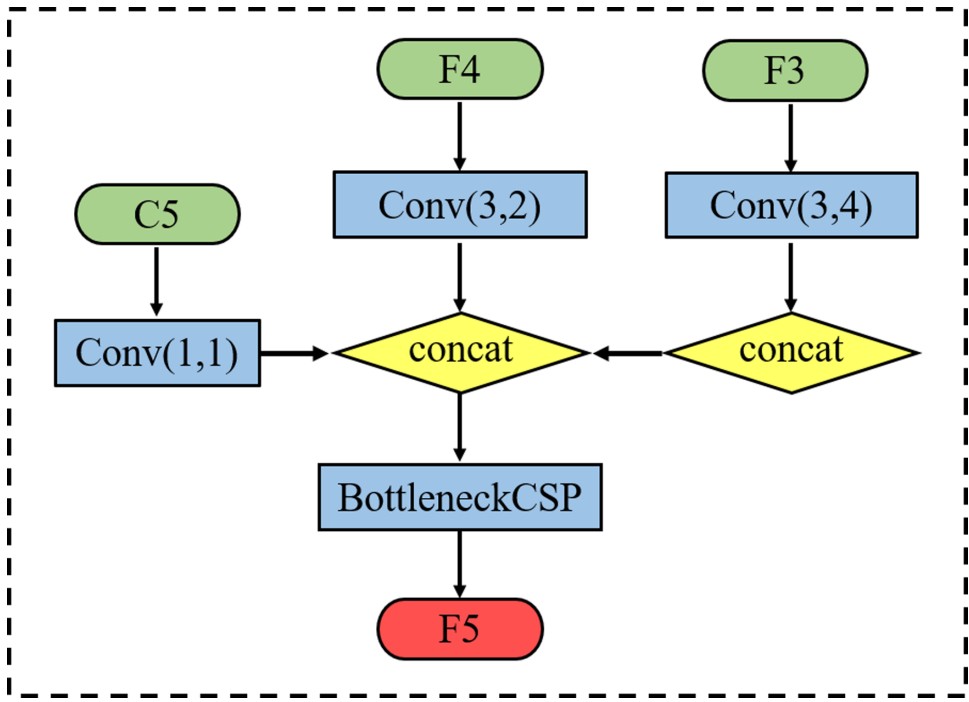

**Figure 3** Schematic diagram of feature P5 acquisition process.

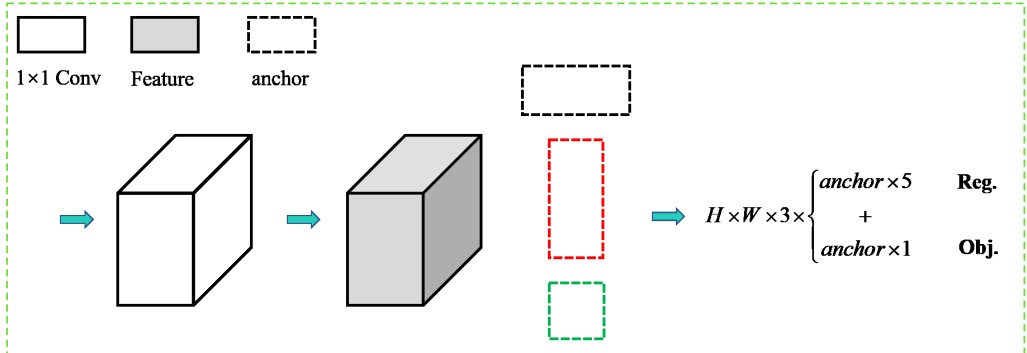

**Figure 4** The output diagram of the rotational decoupled bounding box detection module.

horizontal anchor box desings. Secondly the bounding box in the regression stage of the bounding box prediction is encoded. Finally, the loss function is added to complete the model iteration by the BP algorithm.

## Rotational decoupling bounding box characterization method

Currently, the two main rotational bounding box characterisation methods for rotational characterization arethe eight-parameter method $(x_1, y_1, x_2, y_2, x_3, y_3, x_4, y_4)$ (*Zhu et al., 2015*) and the five-parameter method $(x, y, w, h, \theta)$ (*Ren et al., 2017*). Amongst them, the five-parameter method is divided into two types according to the range of angular

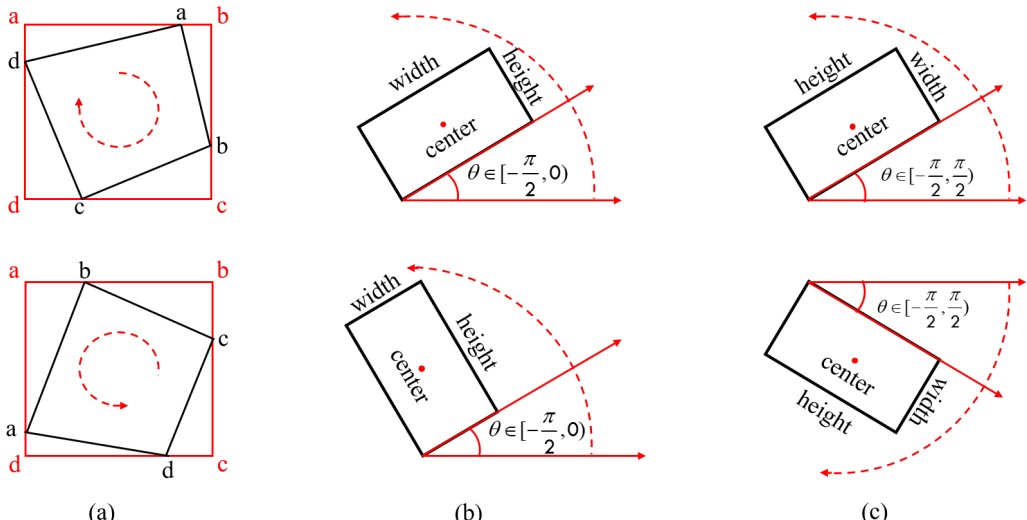

**Figure 5  Common rotational bounding box characterization methods.** (A) Eight-parameter definition method (B) OpenCV definition method (C) Long-edge definition method.

definition variation, namely the OpenCV definition method $\theta \in [-\pi/2, 0)$ and the long-edge definition method $\theta \in [-\pi/2, \pi/2)$, and the specific rotational characterisation is shown in Fig. 5.

All of the characterization methods are capable of detecting directed text, but the eight-parameter characterisation method requires significantly more model parameters than the five-parameter characterization method. Both the OpenCV definition method and the long-edge definition method have issues with angular periodicity, thereby a effecting the stability of model training (*Yang & Yan, 2020*). A new rotationally decoupled bounding box five-parameter characterisation technique, shown in Fig. 6, is suggested in this study.

The rotational decoupling bounding boxes come in two varieties: $w \geq h(|\theta| \leq \pi/4$ with $X$-axis) and $w < h(|\theta| < \pi/4$ with $Y$-axis). The two types of bounding boxes are symmetric and independent of each other. During the model training, the rotational decoupling bounding box is decoupled to the horizontal bounding box $HBB_X + \theta$ when $w \geq h$ and to the horizontal bounding box $HBB_Y + \theta$ when $w < h$. Under this definition $\theta \in [-\pi/4, \pi/4]$, the decoupled bounding box representation is compatible with the original YOLOv5.

## Positive sample matching strategy

In this study, the head module's positive sample acquisition is based on the algorithmic idea that 'high-quality anchors are easier to regress to obtain an accurate prediction bounding box' in YOLOv4, using the positive sample matching strategy based on the maximum IOU threshold, as shown in Fig. 7. Initially, the anchor is matched with the ground truth bounding box to obtain the IOU, and when the IOU is larger than the set threshold (the threshold value is set to 0.7 in this study to ensure the precision of the detection results and avoid the difficulty of convergence of the model training due to the small number of

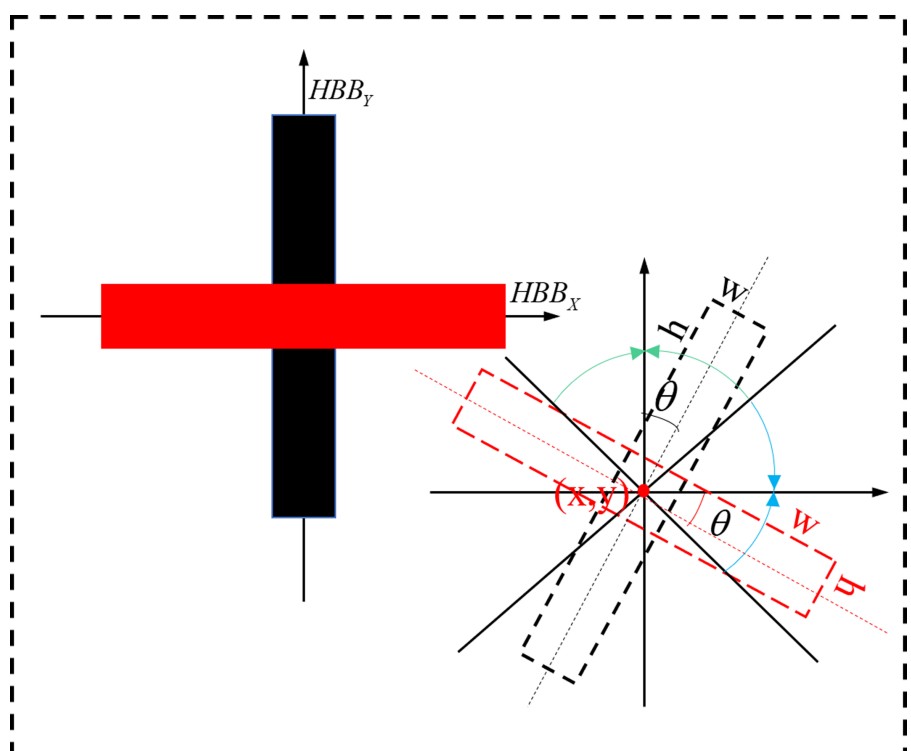

**Figure 6** Schematic representation of the rotational decoupling bounding box.

positive samples), it is marked as a positive sample and mapped to the feature layer, and vice versa for the background.

Regression to obtain precise predicted bounding box is simpler when using a positive sample sampling strategy based on IOU thresholding, but determining how to increase the IOU of matching the ground truth bounding box with the anchor box becomes crucial. The OBB bounding box and HBB anchor box matching cannot achieve a better IOU when the conventional five-parameter bounding box characterisation method is used. The parameter calculation of the model is evidently enhanced by introducing the OBB anchor box, However, the problem of introducing the OBB anchor box to improve the IOU can be effectively avoided by decoupling the OBB bounding box into the combination of $HBB_{X/Y} + \theta$ through the rotation decoupling bounding box characterisation method. As shown in Fig. 8, any rotated bounding box is decoupled into a unique horizontal bounding box $HBB_{X/Y}$ matched with the horizontal anchor box, and the decoupled result $\theta$ is used for loss calculation, allowings the model to calculate the angle between the bounding box and the predicted bounding box, effectively avoiding the problem of angle periodicity in the OpenCV definition method and the long-edge definition method.Moreover, the two positive sample matching strategies of introducing a rotating anchor box and a rotating decoupled bounding box are compared. The result shows that both rotating decoupled bounding boxes can match with an HBB anchor box to obtain a higher IOU and improve the accuracy rate of the prediction bounding box.

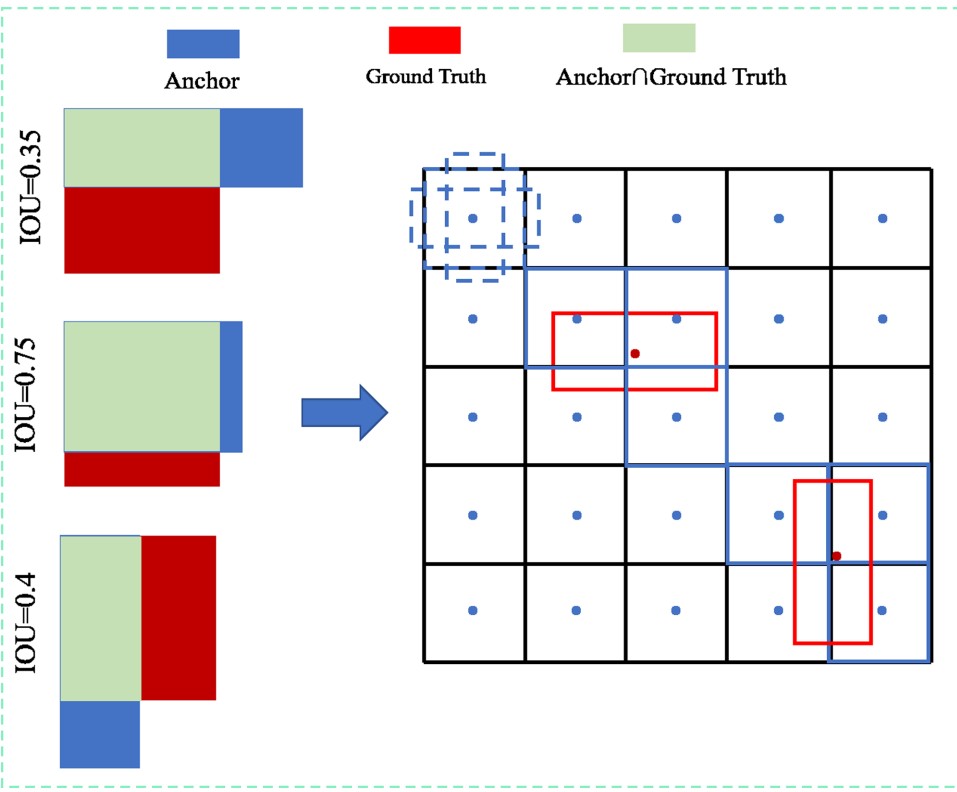

**Figure 7** Schematic diagram of the positive sample matching strategy based on the maximum IOU threshold.

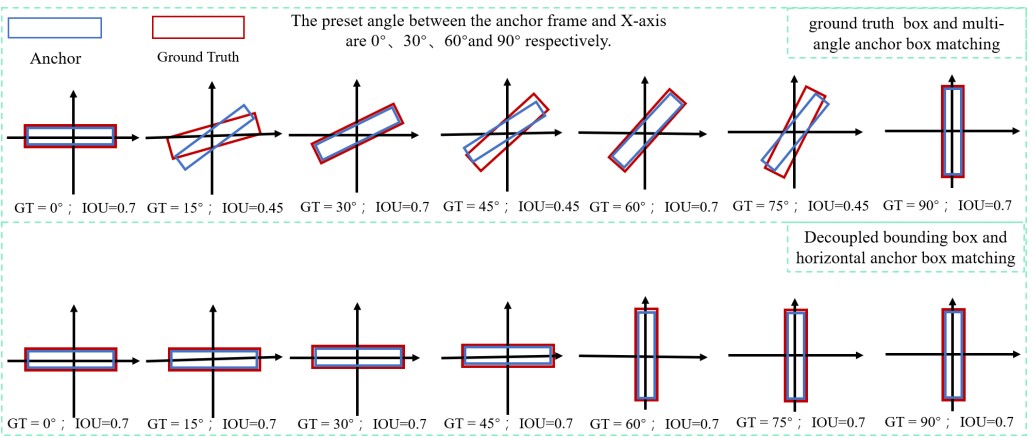

**Figure 8** Schematic diagram of matching the real bounding box with the anchor box before and after decoupling.

## Bounding box coding

After the positive samples are obtained by the matching strategy, they need to be predicted by the bounding box regression. However, in the implementation of the algorithm, if the obtained positive samples are directly subjected to the bounding box regression, then oscillations in the losses occur due to the absence of any constraints on the parameters of the bounding box, which is not conducive to model training. The parameters of the bounding box are encoded to effectively carry out model training. The encoding principle is shown in Fig. 9, where the model initially restricts the regression prediction bounding box to a certain offset $(t_x, t_y, t_w, t_h)$ of the corresponding anchor box by indexing the grid coordinates $(c_x, c_y)$ and presetting the anchor box width and height $(p_w, p_h)$. Then, the output range is controlled to $[-1, 1]$ by using the tanh function according to the angle change interval $\theta \in [-\pi/4, \pi/4]$ in the bounding box, strictly controlling the model's angle prediction interval $t_\theta$, and finally obtaining the prediction angle $(\hat{x}, \hat{y}, \hat{w}, \hat{h}, \hat{\theta})$ by using the angle decoding formula. The encoding of the offset and the decoding of the predicted bounding box are calculated as follows:

$$
\begin{cases}
t_x = x - c_x \\
t_y = y - c_y \\
t_w = \log(w/p_w) \\
t_h = \log(h/p_h) \\
t_\theta = 4\theta/\pi
\end{cases}
\tag{2}
$$

$$
\begin{cases}
\hat{x} = 2\sigma(t_x) - 0.5 + c_x \\
\hat{y} = 2\sigma(t_y) - 0.5 + c_y \\
\hat{w} = p_w \times 2\sigma(t_w)^2 \\
\hat{h} = p_h \times 2\sigma(t_h)^2 \\
\hat{\theta} = \pi/4 \cdot \tanh(t_\theta)
\end{cases}
\tag{3}
$$

where is denoted as a Sigmoid function to scale the predicted bounding box offset to between 0 and 1.

## Loss function

Since the model introduces the OBB bounding box, an angle regression channel needs to be added to this model's bounding box prediction regression channel, and the SmoothL1 loss function is used for angle learning. At the same time, as a single-category text prediction model, it discards the category prediction channel and completes text prediction by OBB prediction regression with a confidence level only to improve the model detection efficiency. The above analysis indicates thatthe loss in this study consists of three types: bounding box loss, angle loss and confidence loss; the specific losses are calculated after modification, as follows:

$$
Loos = loss_{bbox} + loss_{theta} + loss_{conf}
\tag{4}
$$

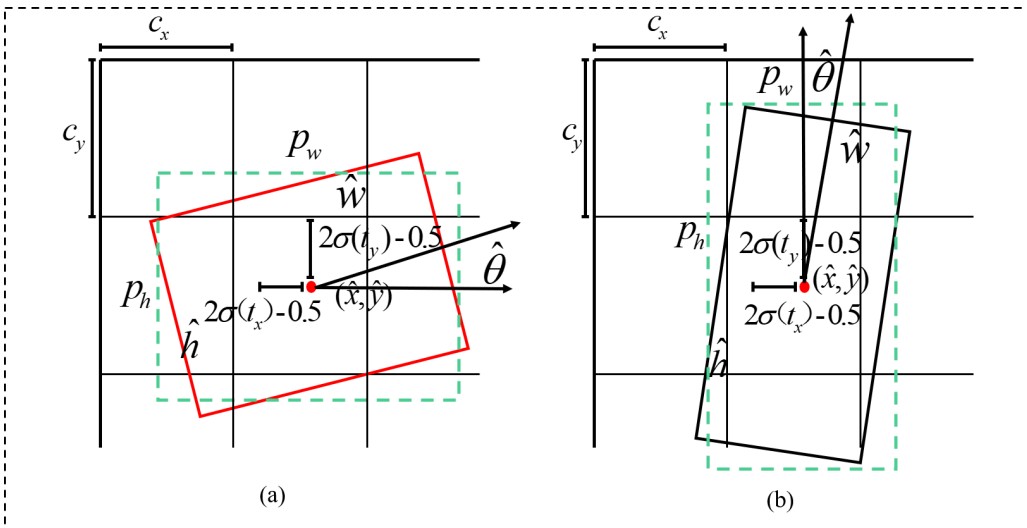

**Figure 9** **Rotationally decoupled bounding box regression schematic.** (A) Pw $\geq$ Ph (B) Pw < Ph.

$$loss_{\text{bbox}} = \lambda_{coord} \sum_{i=0}^{S^2} \sum_{j=0}^{B} I_{ij}^{obj} \left[ (\hat{x}_i^j - x_i^j)^2 + (\hat{y}_i^j - y_i^j)^2 \right] + \lambda_{coord} \sum_{i=0}^{S^2} \sum_{j=0}^{B} I_{ij}^{obj} (2 - \hat{w}_i^j \times \hat{h}_i^j)^2$$
$$\times \left[ (\hat{w}_i^j - w_i^j)^2 + (\hat{h}_i^j - h_i^j)^2 \right] \tag{5}$$

$$loss_{theta} = \lambda_{coord} \sum_{i=0}^{S^2} \sum_{j=0}^{B} I_{ij}^{obj} smooth_{L1}(\hat{\theta}_i^j - \theta_i^j) \tag{6}$$

$$loss_{conf} = \sum_{i=0}^{S^2} \sum_{j=0}^{B} -I_{ij}^{obj} \left[ c_i^j \ln \hat{c}_i^j + (1 - c_i^j)\ln(1 - \hat{c}_i^j) \right] + \lambda_{noobj} \sum_{i=0}^{S^2} \sum_{j=0}^{B} -I_{ij}^{noobj} (c_i^j - \hat{c}_i^j) \tag{7}$$

where $I_{ij}^{obj}$ denotes whether the $i$ th anchor box in the $j$ th grid is responsible for the predicted text bounding width position, and the value of which is 1, and 0 otherwise. $(x_i^j, y_i^j, w_i^j, h_i^j, \theta_i^j, c_i^j)$ and $(\hat{x}_i^j, \hat{y}_i^j, \hat{w}_i^j, \hat{h}_i^j, \hat{\theta}_i^j, \hat{c}_i^j)$ denote the real text marker bounding box and the predicted bounding box parameters and the confidence level, respectively; $\lambda_{coord}$ and $\lambda_{noobj}$ denote the penalty coefficients.

## Rotationally decoupled bounding box detection module implementation

Based on the baseline network design of rotationally decoupled bounding box and horizontal anchor box positive sample matching strategy, predicted bounding box regression coding and loss calculation, the rotationally decoupled bounding box detection module can effectively realize autonomous learning and inference, and its specific learning and inference process are shown in Tables 1 and 2, respectively.

**Table 1  Training process of rotation-based decoupled bounding box detection module.**

**Training process of rotation-based decoupled bounding box detection module**

Input: model prediction channel parameters $\hat{T}$ andreal text marker parameters $T$,

   where: $\hat{T} = (\hat{x}, \hat{y}, \hat{w}, \hat{h}, \hat{\theta}, \hat{P}_r); T = (x, y, w, h, \theta, P_r)$

Output: *Loss*

1. Extract the prediction bounding box parameters and the ground bounding box parameters;

$\hat{T}_{HBB} = (\hat{x}, \hat{y}, \hat{w}, \hat{h});$

$\hat{T}_{theta} = \hat{\theta};$

$\hat{T}_{obj} = \hat{P}_r;$

$T_{HBB} = (x.y, w, h);$

$T_{theta} = \theta;$

$T_{obj} = P_r$

2. Regression coding of bounding box parameters, indexing grid coordinates $(c_x, c_y)$;

Preset anchor width and height $(p_w, p_h)$, Learning regression predicts the corresponding offset of the bounding box to the anchor $(t_x, t_y, t_w, t_h, t_\theta)$. where: $t_x = x - c_x;$

$t_y = y - c_y;$

$t_w = \log(w/p_w);$

$t_h = \log(h/p_h);$

$t_\theta = 4\theta/\pi$

3. Calculate the loss of each parameter;

$Loss_{HBB} = L_{CIOU}(\hat{T}_{HBB}, T_{HBB});$

$Loss_{theta} = L_{smoothL1}(\hat{T}_{theta}, T_{theta});$

$Loss_{HBB} = L_{BCE}(\hat{T}_{obj}, T_{obj})$

**Table 2  Inference process of rotationally decoupled bounding box detection module.**

**Inference process of rotationally decoupled bounding box detection module**

Input: The model prediction channel predicts the corresponding offset of the bounding box to the anchor box $(t_x, t_y, t_w, t_h, t_\theta)$

Output: Prediction bounding box $\hat{T} = (\hat{x}, \hat{y}, \hat{w}, \hat{h}, \hat{\theta}, conf)$

1. Decode the offset $(t_x, t_y, t_w, t_h, t_\theta)$ of the predicted bounding box of the regression channel with respect to the anchor box, Get the prediction bounding box parameters: $\hat{x} = 2\sigma(t_x) - 0.5 + c_x;$

$\hat{y} = 2\sigma(t_y) - 0.5 + c_y;$

$\hat{w} = p_w \times 2\sigma(t_w)^2;$

$\hat{h} = p_h \times 2\sigma(t_h)^2$

2. Obtain the confidence level of the text location target and filter the confidence level threshold to obtain the prediction result $\hat{T} = (\hat{x}, \hat{y}, \hat{w}, \hat{h}, \hat{\theta}, \hat{P}_r);$

3. Non-Maximum Suppression processing of the acquisition results $\rightarrow NMS_{OBB}(\hat{T})$

4. Output the final prediction bounding box $\hat{T} = (\hat{x}, \hat{y}, \hat{w}, \hat{h}, \hat{\theta}, conf)$

## Analysis of experimental results

In this section, we first present the experimental data and experimental setup of the proposed method, then analyse the model performance of the proposed method by conducting comparison tests on a public dataset, and finally analyse the effectiveness of different improved methods by ablation experiments on a self-built dataset.

**Table 3  Deep learning environment configuration table.**

| Platform | CPU | GPU | Memory | Running environment configuration | | | |
|----------|-----|-----|--------|--------|--------|--------|--------|
| PC | Intel i7-8700 | Nvidia Pascal1060 6G | 16G | CUDA9.0 | cudnn7.1 | PyTorch1.8 | Python3.8 |

## Experimental data and experimental setup

For this experiment, the model was evaluated on ICDAR 2015, MSRA-TD 500 and a self-built dataset to test the model's performance. In addition, to shorten the experimental data training and its validation time, training and testing were performed on a 64-bit Windows system with the hardware configuration and running environment configuration shown in Table 3. During model training, the model used stochastic gradient descent as the optimiser, the momentum factor was set to 0.9, the weight decay was set to 0.005, the initial weights of the network model were generated by random initialisation, the initial learning rate was 0.001, 16 images were input in each iteration, and the learning rate was multiplied by 0.1 at 100 and 200 times to reduce the learning rate, and the regularisation factor was set to 0.0005 to suppress overfitting. The coefficient of regularisationwas set to 0.0005 to suppress overfitting.A total of 300 iterations were performed, the model was saved every one iteration, and the best weight amongst 300 iterations was finally selected as the detection network model.

## Model comparison experiments

To verify the effectiveness of the proposed directed text detection model based on rotation decoupled bounding boxes, it compared and analysed by validating it on the MSRA-TD500 and ICPR2015 datasets with recent years text detection algorithms. The recent years algorithms mainly include CTPN (*Tian et al., 2016*), RRPN (*Ma et al., 2018*), R2CNN (*Jiang et al., 2017*), EAST (*Zhou et al., 2017*), DBNet (*Liao et al., 2020*), SegLink (*Shi, Bai & Belongie, 2017*), TextSnake (*Long et al., 2018*), PSENet (*Wang et al., 2019a*) and PANNet (*Wang et al., 2019b*), ATRR (*Wang et al., 2019c*), TextFuseNet (*Ye et al., 2020*). The model's performance is referenced by four metrics: *precision*, *recall*, $F_1$-*Score*, and detection speed, which are calculated as follows:

$$Precision = \frac{TP}{TP + FP} \tag{8}$$

$$Recall = \frac{TP}{TP + FN} \tag{9}$$

$$F1 - Score = \frac{2 \times Precision \times Recall}{Precision + Recall} \tag{10}$$

where $TP$ is the positive sample whose model prediction is accurate, $FP$ is the negative sample whose model prediction is positive, and $FN$ is the positive sample whose model prediction is inaccurate.

Table 4 shows the performance results of different algorithm models on the MSRA-TD500 dataset, where a segmentation-based DBNet algorithm achieves an accuracy of

**Table 4** Detection results of the algorithms in this article and those in recent years on the MSRA-TD500 dataset.

| Methods | P (%) | R (%) | F (%) | Speed (fps) |
|---|---|---|---|---|
| RRPN | 82.0 | 73.0 | 77.0 | 3.3 |
| EAST | 87.3 | 67.4 | 76.1 | 13.2 |
| DBNet | 91.5 | 79.2 | 84.9 | 32 |
| SegLink | 86.0 | 70.0 | 77.0 | 8.9 |
| TexSnake | 83.2 | 73.9 | 78.3 | 1.1 |
| ATRR | 85.2 | 82.1 | 83.6 | 10 |
| PANNet | 84.4 | 83.8 | 84.1 | 30.2 |
| OUR | 89.2 | 87.0 | 88.1 | 42 |

**Notes.**
In addition to our algorithm for the experimental measurement of the rest of the algorithm detection results are the original algorithm author experimental results.

**Table 5** Detection results of the algorithms in this article and those in recent years on the ICDAR2015 dataset.

| Methods | P (%) | R (%) | F (%) | Speed (fps) |
|---|---|---|---|---|
| CTPN | 74.0 | 52.0 | 61.0 | 7.1 |
| R2CNN | 85.6 | 79.7 | 82.5 | – |
| DBNet | 91.8 | 83.2 | 87.3 | 12 |
| SegLink | 73.1 | 76.8 | 75.0 | – |
| TextFuseNet | 91.3 | 88.9 | 90.0 | 8.3 |
| PSENet | 86.9 | 84.5 | 85.7 | 1.6 |
| PANNet | 84.0 | 81.9 | 82.9 | 26.1 |
| OUR | 90.6 | 88.1 | 89.3 | 39.7 |

**Notes.**
In addition to our algorithm for the experimental measurement of the rest of the algorithm detection results are the original algorithm author experimental results.

91.5 in the dataset detection compared with the comparison algorithm, which achieves the best performance in terms of detection accuracy. However, the proposed rotational decoupled bounding box-based directed text detection algorithm model is second only to the DBNet algorithm in terms of accuracy and achieves better performance results in terms of recall, $F_1$-Score, and detection speed compared with the other algorithms. Table 5 shows the performance results of different algorithmic models on the ICDAR2015 dataset, in which the frame rate measured by the proposed model is 39.7 fps, and its detection speed achieves the best amongst other algorithms. In addition, the detection *precision*, *recall* and $F_1$-Score of the proposed model are similar to the SOTA methods, effectively verifying that the proposed model is competitive. Figure 10 displays some of the detection outcomes of the proposed arithmetic model used in this article on the MSRA-TD500 and ICDAR2015 datasets. It is clear from the figure that the model is capable of performing directed text detection.

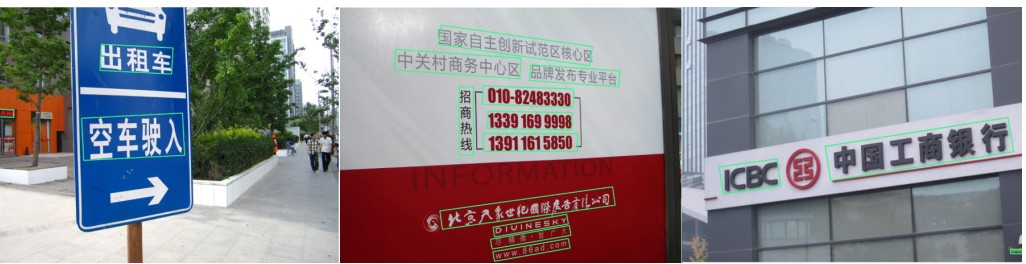

**Figure 10** Partial detection results of public dataset.

**Table 6** Effect of the performance of the D-PANet feature fusion module.

| Methods | P (%) | R (%) | F (%) | Speed (fps) |
|---|---|---|---|---|
| PANet | 60.9 | 58.3 | 59.6 | 42.1 |
| D-PANet | 62.1 | 58.6 | 60.1 | 41.2 |

## Ablation experiments

The improved YOLOv5 multiscale special fusion sign pyramid D-PANet and multi-angle rotation decoupled bounding box detection modules on self-built datasets are tested and compared under the same deep learning environment and parameter configuration to analyse the effectiveness of different improved methods and to use the proposed methods in an industrial real-world environment.

(1) Effectiveness of D-PANet feature fusion module.

Table 6 demonstrates the effect of whether the model employs the D-PANet feature fusion module on the performance of the detection network. The experimental results show that the use of the D-PANet feature fusion module leads to a 1.2%, 0.3% and 0.5% improvement in the model detection accuracy, recall, and F-value, respectively, compared with the original model performance, and the detection speed only decreases by 0.9 fps, effectively demonstratingthat the D-PANet dual-branch feature information fusion structure can further improve the performance of the shallow and deep feature information fusion and highlight the target information, thereby improving text detection precision with only a small increase in parameter calculation.

(2) Effectiveness of rotating decoupled bounding box detection modules

Four experimental control groups are designed in this experiment to investigate the effect of different bounding box detection modules on the performance of the detection network. The first group of experiments uses the original YOLOv5 detection module; the second group of experiments uses the OpenCV bounding box definition method detection module; the third group of experiments uses the long-edge definition method detection module; the fourth group uses the decoupled bounding box detection module. The performance of the detection models is verified by experiments, as shown in Table 7.

The comparison betweenthe first group and the fourth groupindicates that the detection speed of the decoupled bounding box detection module is reduced by 3.6 fps, but the detection speed is improved by 6.7 and 6.5 fps compared with the second group and the

**Table 7  Effect of different boundary frame characterizations on the performance of the detection module.**

| Experimental group | OpenCV definition method | Long-edge definition method | Rotational decoupling | R (%) | P (%) | F (%) | Speed |
|---|---|---|---|---|---|---|---|
| Group 1 | | | | 60.9 | 58.3 | 59.6 | 42.1fps |
| Group 2 | ✓ | | | 90.4 | 87.8 | 89.1 | 30.8fps |
| Group 3 | | ✓ | | 92.6 | 89.4 | 90.8 | 31.0 fps |
| Group 4 | | | ✓ | 95.6 | 90.1 | 92.8 | 37.5 fps |

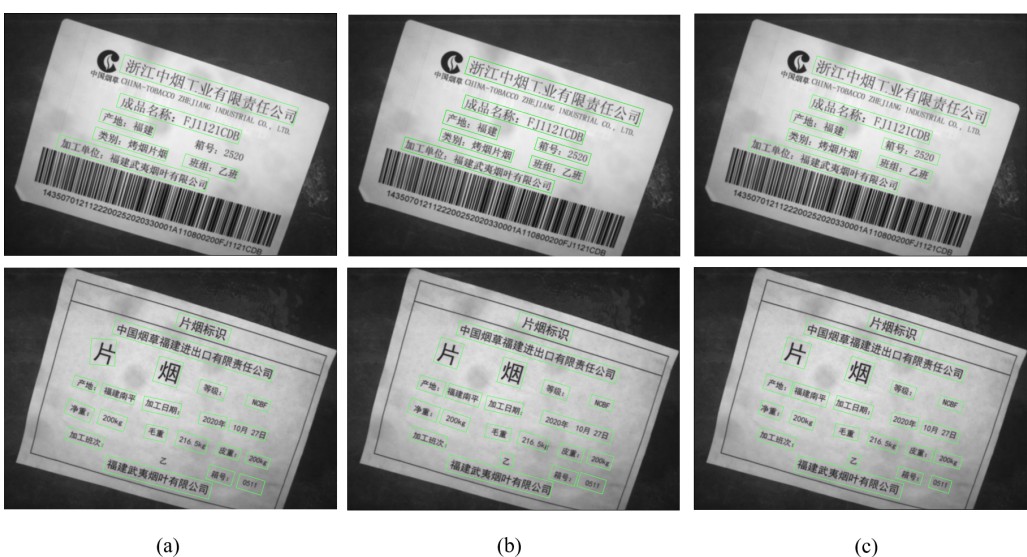

(a)                            (b)                            (c)

**Figure 11  Partial model test plots of the detection module for different bounding box representations.**
(A) OpenCV definition method detection module; (B) Long-edge definition method detection module;
(C) Rotationally decoupled bounding box detection module.

third group, respectively. This finding shows that the decoupled bounding box detection module can effectively avoid the introduction of rotating anchor boxes to reduce the model parameter calculation and improve the model detection efficiency. Meanwhile, the model detection precision, recall and $F_1$-score, are 95.6%, 90.1%, and 82.8%, respectively, using the decoupled bounding box detection module. The detection performance is improved compared with the second and third groups, effectively showingthat the decoupled bounding box detection module can solve the problem of angular periodicity in the regression bounding box and improve the model detection performance. The results of the corresponding detection model are shown in Fig. 11.

## CONCLUSION

In this study, a rotation-based decoupled directed text detection algorithm is proposed. The algorithm is built by the YOLOv5 model framework. The target feature information fusion is enhanced by a two-branch feature fusion structure. Then, a rotationally decoupled bounding box representation is defined, and a positive sample matching strategy between

the rotationally decoupled bounding box and the horizontal anchor box is proposed to improve the data positive sample acquisition capability. Subsequently, the bounding box is encoded, and the corresponding loss function is added in the predictive regression stage of the bounding box to complete the construction of the rotationally decoupled bounding box detection module and improve the performance of the directed text detection. Comparative experiments are conducted on several rotation detection datasets, such as MSRA-TD500 and ICDAR2015,to verify the detection performance of the model proposed. The experimental results show that the proposed method in this article achieves better detection accuracy and efficiency.

Since the continuous development of deep learning techniques, the current detection methods should be improved using different architectures and various text detection methods. In addition, this work focuses on improving the detection accuracy of text with multiple sources, dense distribution, large aspect ratio and arbitrary alignment direction by rotating the decoupled bounding box detection module; thus, the model feature extraction capability is neglected. Although a two-branch feature fusion structure is proposed to enhance the fusion of target feature information in the feature fusion stage, it does not change the problem of insufficient model feature extraction capability. For this reason, the model performance should be further improved in the future work with the help of the previous research base, such as through the YOLOv8 framework and using 2*2 convolutional layers.

### Funding

This project has been supported by the National Natural Science Foundation (No. 52275413), the Natural Science Foundation of Fujian Province (No. 2020J01874), the Program for Innovative Research Team in Science and Technology in Fujian Province University (No. 12), the Fujian Provincial Key Project of Science and Technology Innovation (No. 2022G02007) and the High-level talents foundation of Fuzhou Polytechnic (No. FZYRCQD 201903). The funders had no role in study design, data collection and analysis, decision to publish, or preparation of the manuscript.

### Grant Disclosures

The following grant information was disclosed by the authors:
The National Natural Science Foundation: 52275413.
the Natural Science Foundation of Fujian Province: 2020J01874.
The Program for Innovative Research Team in Science and Technology in Fujian Province University: (No, 12).
Fujian Provincial Key Project of Science and Technology Innovation: 2022G02007.
High-level talents foundation of Fuzhou Polytechnic: FZYRCQD 201903.

### Competing Interests

Minrui Lu and Tengjian Zhang are employed by Fujian Wuyi Leaf Tobacco Co., Ltd.

## Author Contributions

- Songma Wei conceived and designed the experiments, performed the experiments, analyzed the data, performed the computation work, authored or reviewed drafts of the article, and approved the final draft.
- Minrui Lu conceived and designed the experiments, analyzed the data, authored or reviewed drafts of the article, and approved the final draft.
- Bingsan Chen conceived and designed the experiments, analyzed the data, authored or reviewed drafts of the article, and approved the final draft.
- Tengjian Zhang analyzed the data, prepared figures and/or tables, and approved the final draft.
- Fujiang Zhang analyzed the data, prepared figures and/or tables, and approved the final draft.
- Xiaodong Peng analyzed the data, prepared figures and/or tables, and approved the final draft.

## Data Availability

The data are available at:

- The MSRA Text Detection 500 dataset (MSRA-TD500):

http://www.iapr-tc11.org/mediawiki/index.php/MSRA_Text_Detection_500_Database_%28MSRA-TD500%29

- ICDAR2015 dataset: https://rrc.cvc.uab.es/?ch=4&com=tasks

The datasets are available at figshare:

- Repartition of part of MSRA-TD500 dataset: Wei, Songma (2023): MSRA-TD500.zip. figshare. Figure. https://doi.org/10.6084/m9.figshare.22323595.v1

- Repartition of part of ICDAR2015 dataset: Wei, Songma (2023): icdar2015.zip. figshare. Figure. https://doi.org/10.6084/m9.figshare.22323535.v1

The code is available in the Supplemental File.

## Supplemental Information

Supplemental information for this article can be found online at http://dx.doi.org/10.7717/peerj-cs.1352#supplemental-information.

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
