# Peer review of "Efficient algorithm for directed text detection based on rotation decoupled bounding box"

_PeerJ Computer Science, doi:10.7717/peerj-cs.1352_

## Round 0.1 · original submission · Minor Revisions

The manuscript is well organized. Based on the reviewer comments, please revise the manuscript and resubmit.

Reviewer 1 ·

Basic reporting

The article is well-written, the literature analysis is adequate, the knowledge gap is well-identified, the structure is linear, and the methodology is well-described, and clearly presented. The arguments are strong and fully documented, and the results are consistent and backed by several metrics and experiments.
In addition, some "micro" concerns attracted the reviewer's attention:
- Line 39. To improve the clarity of the text, it would be appropriate to make the name of the metric "Intersection Over Union" explicit before the abbreviation "IOU" the first time it is mentioned, as has been done for the other terms.
- Lines 40-41. It would be beneficial to include the literature reference(s) for "some scholars" mentioned to increase the text's reliability.
- Lines 52-61. A quick summary of the paper's organization, introducing the major sections, could be added after this paragraph to enable the reader to acquire a clearer picture of the content. As an example, "The paper is organized as follows: Section 2 outlines the approach..., Section 3..." and so on.
- Lines 69-70. The four parts of the architecture (input layer, feature extraction backbone network, feature fusion module, and detection output module) should be made explicit and underlined in Fig.1, in my opinion, to ensure that more readers could easily understand it.
- Line 119. There may be a typo in this line. Did the authors intend to cite any specific works when they wrote: “[5]”?
- Line 132: It would be helpful to explain how the indicated threshold was set to make the description more precise.
- Lines 183-184. I believe that something is missing from this phrase because it is difficult to understand. Could you please clarify the sentence?
- Line 186. There is a typo in this line: table1 and 2 -> Table 1 and 2.
- Lines 222-228. This sentence (From "Table 5 shows..." to "... the strong competitiveness of this model", in my opinion, is excessively wordy and difficult to understand. To make it more comprehensible and readable, I would propose rephrasing it.
- Lines 270-281. Is there still room for improvement? If so, is there a parameter that the authors would concentrate on? I would advise including cues for possible upgrades or future works.

Experimental design

No additional comments.

Validity of the findings

No additional comments.

Additional comments

No additional comments.

Reviewer 2 ·

Basic reporting

The authors applied their proposed model to two well-known benchmark datasets. The paper is generally well-written and well-structured. Related work is thorough.

Experimental design

The paper utilizes a YOLO detection-based framework as a backbone for scene text detection. They combined this framework with the proposed D-PANet feature fusion network which is their main contribution to this paper. The combination is trivial and it leads to good performance later.

Validity of the findings

Experimental results are compelling: state-of-the-art on MSRA-TD500 has rotated long text.
It also achieved good frame-per-second efficiency on ICDAR15 contains high-resolution images.

Additional comments

I think the paper could also be made more comprehensive with an analysis of where the model does fail or experiments on other datasets/different environments.

·

Basic reporting

The paper is well researched and very well written. The authors have performed extensive research and benchmarked results on several neural network models. The authors have also gone deep into the library provided by OpenCV. They explained about the practicality that it may be a bit slow but very accurate. They have gone in detail to explain about training the network - time, iterations, dataset size etc. Generally when it comes to detection, the text images are corrected horizontally before feeding it to the model. Some of the images show bounding boxes at angles and can detect images in the wild with angles. This is a great solution.

Experimental design

Maybe the authors might have already performed enough research before the launch of YOLO V8 which is incredibly fast. Maybe the authors can try using YOLO V8 architecture concept and see if it helps in imporving the accuracy and speed.

Validity of the findings

No Comments

Reviewer 4 ·

Basic reporting

Review for paper – An efficient algorithm for directed text detection based on rotation decoupled bounding box.

Abstract – Introduction is well framed, and problem is stated. I would suggest to add references for this statement ‘the problem of low accuracy in detecting text’. The datasets are mentioned with the results of the model. Accuracy is typically not the best metric in classification problems. Mention it to confusion matrix and F1 scores along with the accuracy.

Introduction – References are correctly mentioned, and historical context is provided aptly. I would suggest adding research happened in recent times around 2021/2022. What methods does other researchers used for improving rotational decoupling border and what other recent architectures are used for detection.

Experimental design

Methods/Model/Data – Proposed method is mentioned very precisely with all architecture diagrams and formulas. D-PANET feature Fusion network is used in the right context and explained very neatly with formulas. I would advise to try with 2*2 convolutional layer which has better performance in recent times with detection problems. Also fine tune the size of the layer to get the best fit. Boundary frame characterization methods are mentioned clearly with references and calculation of bounding box encoding is right. Loss function also aptly described for tis problem. Advise to label the formula more clearly and there are some grammatical errors in proposed method paragraphs.

Validity of the findings

Results – Results are correctly measured and relevant with the problem. Choosing F1-score, precision and recall is correct in this cases.

Conclusion – Expand more on conclusion. What can be done more in future in this field and problem.

---

## Round 0.2 · accepted · Accept

Based on the reviewer comments I am provisionally accepting the manuscript for publication.

Reviewer 1 ·

Basic reporting

no comment

Experimental design

no comment

Validity of the findings

no comment

Additional comments

no comment

Reviewer 4 ·

Basic reporting

Revisions are done correctly.

Experimental design

Revisions are done correctly.

Validity of the findings

Revisions are done correctly.